# Comprehensive Analysis Identifies *THEMIS2* as a Potential Prognostic and Immunological Biomarker in Glioblastoma

**DOI:** 10.3390/cells14020066

**Published:** 2025-01-07

**Authors:** Jianan Chen, Qiong Wu, Anders E. Berglund, Robert J. Macaulay, Arnold B. Etame

**Affiliations:** 1Department of Neuro-Oncology, H. Lee Moffitt Cancer Center and Research Institute, Tampa, FL 33612, USA; jianan.chen@moffitt.org (J.C.); qiong.wu@moffitt.org (Q.W.); 2Department of Biostatistics and Bioinformatics, H. Lee Moffitt Cancer Center and Research Institute, 12902 Magnolia Drive, Tampa, FL 33612, USA; anders.berglund@moffitt.org; 3Departments of Anatomic Pathology, H. Lee Moffitt Cancer Center and Research Institute, 12902 Magnolia Drive, Tampa, FL 33612, USA; robert.macaulay@moffitt.org

**Keywords:** macrophage-mediated immunosuppression, glioblastoma, *THEMIS2*, tumor microenvironment, prognosis

## Abstract

Glioblastoma (GBM) is a highly aggressive brain tumor characterized by its ability to evade the immune system, hindering the efficacy of current immunotherapies. Recent research has highlighted the important role of immunosuppressive macrophages in the tumor microenvironment (TME) in driving this immune evasion. In this study, we are the first to identify *THEMIS2* as a key regulator of tumor-associated macrophage (TAM)-mediated immunosuppression in GBM. We found that a high *THEMIS2* expression is associated with poor patient outcomes and increased infiltration of immune cells, particularly macrophages. Functional analyses revealed *THEMIS2*’s critical involvement in immune-related pathways, including immune response activation, mononuclear cell differentiation, and the positive regulation of cytokine production. Additionally, single-cell RNA sequencing data demonstrated that macrophages with a high *THEMIS2* expression were associated with increased phagocytosis, immune suppression, and enhanced tumor growth. These findings suggest that *THEMIS2* could serve as both a prognostic marker and a therapeutic target for enhancing anti-tumor immunity in GBM.

## 1. Introduction

Glioblastoma (GBM) is the most aggressive and prevalent primary brain tumor in adults, characterized by a devastating prognosis [1]. Despite advances in standard treatments, including surgery, radiation, and chemotherapy, the median survival for GBM patients remains only 12–15 months, with a five-year survival rate of less than 5% [2,3]. The inherent heterogeneity and resistance to the current treatments underscore the urgent need for novel diagnostic biomarkers and therapeutic strategies [4].

Tumor-associated macrophages (TAMs) represent a major component of the GBM tumor microenvironment (TME), often exhibiting high expression of co-inhibitory receptors that contribute to immunosuppression [5,6]. Studies have identified inhibitory receptors, such as LILRB3 and SIGLEC9, as independent negative prognostic factors in GBM patients, with higher expression correlating with poorer survival outcomes [7,8]. These findings highlight the crucial role of TAMs and their surface receptors in GBM progression and patient prognosis.

While immunotherapies, such as immune checkpoint blockade (ICB), have achieved therapeutic benefits in several cancers, their efficacy in GBM has been notably limited [9,10]. Several factors contribute to this limited success, including the immunosuppressive microenvironment shaped by myeloid cells like TAMs, lymphopenia resulting from radio- and chemotherapy, and the sequestration of T cells in the bone marrow [11]. Therefore, a deeper understanding of the immunological landscape of GBM, particularly the role of TAMs, is essential for developing effective immunotherapeutic strategies.

THEMIS2 (Thymocyte Selection-Associated Family Member 2) is a protein initially recognized for its role in T-cell development and signaling, with emerging evidence suggesting its significant involvement in immune cell activation and signal transduction [12,13]. However, its role in tumor-associated macrophages (TAMs) and the immunosuppressive environment of GBM remains largely unexplored. Our study is the first to investigate *THEMIS2* in the context of GBM, particularly its role in TAM-mediated immunosuppression.

Here, we aim to investigate *THEMIS2* expression in GBM and its association with clinicopathological features and patient prognosis. Using data from The Cancer Genome Atlas (TCGA) and the Chinese Glioma Genome Atlas (CGGA), we assess the expression levels of *THEMIS2* in macrophages and their correlation with overall survival in GBM patients. Our findings intend to provide insights into the molecular mechanisms of GBM and contribute to the development of new therapeutic targets.

## 2. Materials and Methods

### 2.1. Patients and Samples

Publicly available gene expression datasets of patients with glioblastoma (GBM) were obtained from The Cancer Genome Atlas (TCGA, https://cancergenome.nih.gov/, accessed on 15 August 2024) and the Chinese Glioma Genome Atlas (CGGA, http://www.cgga.org.cn/, accessed on 15 August 2024). In addition to bulk RNA-seq data from TCGA-GBM and CGGA RNAseq datasets (CGGA325) [14], single-cell RNA sequencing (scRNA-seq) data from CGGA were also utilized to further investigate the cellular heterogeneity of GBM [15]. Standard preprocessing steps, including quality control, normalization, and clustering, were applied to the single-cell data. All the datasets included clinical and molecular information, such as gender, age, overall survival (OS), survival status, and subtype. Only primary cases with complete survival data were retained for analysis.

### 2.2. Gene Expression Analysis

We used the GEPIA tool (http://gepia.cancer-pku.cn, accessed on 20 August 2024) to analyze *THEMIS2* gene expression based on the TCGA and GTEx data, focusing on low-grade glioma (LGG) and GBM. RNA sequencing data from TCGA were also used to examine *THEMIS2* expression across clinical subgroups, including molecular subtypes, MGMT promoter methylation status, and age groups. Statistical analyses were performed using the R software (version 4.4.0), with *p*-values less than 0.05 considered statistically significant. Box plots were generated using ‘ggplot2’ (v3.5.1) to visualize the distribution of *THEMIS2* expression, with significance levels indicated as * (*p* < 0.05), ** (*p* < 0.01), and *** (*p* < 0.001). Kaplan–Meier survival curves were generated, and statistical significance was assessed using the log-rank test. Survival curves and risk tables were plotted using the ‘survival’ (v3.7.0) and ‘survminer’ (v0.5.0) packages.

### 2.3. Functional Enrichment Analysis

To explore the biological functions and pathways associated with *THEMIS2*-correlated genes, the Gene Ontology (GO), Kyoto Encyclopedia of Genes and Genomes (KEGG) [16], and Gene Set Enrichment Analysis (GSEA) [17] analyses were performed. *THEMIS2*-correlated genes were identified using the RNA sequencing data from the mentioned two databases, selecting the top 500 positively correlated genes with a *p*-value < 0.05 to identify the key genes most closely associated with *THEMIS2*. The GO and KEGG enrichment analyses were conducted using the ‘clusterProfiler’ package (v4.12.6), with the results visualized as bubble and bar plots. The Benjamani–Hochberg method was applied to adjust the *p*-values (*p*.adjust), and terms with *p*.adjust < 0.05 were considered significantly enriched. For GSEA, gene sets from the MSigDB Hallmark collection were used, and enrichment scores (ES) were calculated for each sample. The normalized enrichment score (NES) ranked pathways, with significance assessed using a false discovery rate (FDR) < 0.25. The results were displayed as bar plots. The color gradient represented the negative logarithm of the *p*-value (−log10(pval)) multiplied by the sign of the enrichment score, highlighting both the positive and negative enrichment.

### 2.4. Immune Characteristics Analysis

We analyzed the association between *THEMIS2* expression and immune cell infiltration, tumor purity, and specific immune cells such as macrophages and dendritic cells using TIMER 2.0 (http://timer.cistrome.org/, accessed on 20 August 2024) and visualized the results in scatter plots. Additionally, we used the MCP-counter [18] and xCell [19] algorithms via the “MCPcounter” and “xCell” R packages to further investigate immune cell infiltration, with heatmaps generated to visualize differences in infiltration. GSEA was performed using the “GSVA” [20] package (v1.52.3) to explore the immune-related pathways associated with *THEMIS2* expression. Spearman’s correlation was applied to assess the relationship between *THEMIS2* expression and key immune checkpoint molecules, including PDCD1, CD274, CTLA4, HAVCR2, TIGIT, PDCD1LG2, VSIR, CSF1R, TREM2, IL10, IL10RA, CD163, and SIGLEC15. Correlation heatmaps were generated for both the CGGA and TCGA datasets. Finally, the correlation between *THEMIS2* expression and ESTIMATE scores [21] (StromalScore, ImmuneScore, and ESTIMATEScore) was assessed using the ESTIMATE algorithm. The scatter plots were generated using “ggplot2”.

### 2.5. Single-Cell Transcriptome Analysis

scRNA-seq data were obtained from CGGA, focusing on GBM tumor samples. The gene expression matrices were processed using the Seurat R package [22]. Low-quality cells were filtered out during the preprocessing stage to ensure the accuracy and reliability of the analysis. Specifically, only the genes expressed in at least three cells were retained, and the cells with a detectable gene count ranging from 200 to 7000 were considered high quality and included for further analysis. Additionally, the cells with mitochondrial gene content exceeding 10% were excluded to avoid potential bias caused by stressed or dying cells. After applying these rigorous quality control criteria, the initial dataset, which consisted of 4193 cells, was reduced to 3718 high-quality cells suitable for a downstream analysis. This filtering process ensured the robustness of our results and minimized the inclusion of artifacts or noise.

Following filtering, a principal component analysis (PCA) was performed, followed by clustering using the “FindNeighbors” and “FindClusters” functions in Seurat. Uniform Manifold Approximation and Projection (UMAP) was used to visualize the clustering results based on the top 12 significant principal components. The cells were clustered with a resolution of 14, and cell types were annotated based on the known marker genes from previous studies [23].

### 2.6. Macrophage Subtype Analysis

Macrophages were isolated from the dataset, yielding a total of 1051 cells. These macrophages were grouped into seven major clusters based on their transcriptional profiles using the clustering approach implemented in Seurat. To investigate the role of *THEMIS2*, the macrophages were further divided into high and low *THEMIS2* expression subgroups based on the median expression levels of *THEMIS2*.

A differential expression analysis was conducted using the ‘FindMarkers’ function in Seurat (v5.1.0), comparing the high and low *THEMIS2* expression groups. The genes with a log2 fold change greater than 0.5 and an adjusted *p*-value < 0.05 were identified as significantly differentially expressed. The results were visualized using volcano plots generated with the “EnhancedVolcano” package in R, which highlighted the log2 fold changes and −log10 adjusted *p*-values of the differentially expressed genes. To further explore the biological implications of the identified genes, a Gene Ontology analysis was performed on those with a *p*-value < 0.05. The “cnetplot” function from the clusterProfiler package was used to visualize the network relationships between the genes and their associated enriched pathways, providing insight into the functional roles of *THEMIS2* in macrophages and its potential impact on the tumor microenvironment. These analyses revealed the key pathways and genes that may underlie the differential behavior of macrophages based on *THEMIS2* expression.

### 2.7. Intercellular Communication Analysis

We used the “CellChat” [23] package (v2.1.2) to investigate intercellular communication among ‘Macrophage_HIGH_*THEMIS2*’, ‘Macrophage_LOW_*THEMIS2*’, ‘Malignant’, ‘Oligodendrocyte’, and ‘T-cell’ populations. RNA-sequencing data from the Seurat object identified overexpressed genes and ligand–receptor interactions. Communication probabilities were calculated and aggregated at the signaling pathway level. Circle plots were generated to visualize the number and strength of the interactions between cell types. Subsequently, ligand–receptor pairs were extracted for further analysis. We selected the top 20 ligand–receptor pairs based on communication probabilities, focusing on macrophages with a high *THEMIS2* expression. These pairs were further analyzed, and their functions were enriched in key biological processes, including immune cell regulation, immune escape, migration and invasion promotion, immune suppression, and tumor growth enhancement. A Sankey diagram, generated with RAWGraphs (https://www.rawgraphs.io/, accessed on 29 August 2024), was used to illustrate the connections between cell types, signaling pathways, and their biological functions.

### 2.8. Statistical Analysis

Student’s *t*-test or Wilcoxon rank sum test was used for two-group comparisons, and two-way ANOVA for comparisons among more than two groups. Spearman correlation assessed variable correlations. A Kaplan–Meier analysis was used to evaluate survival rates. The Cox proportional hazard model estimated hazard ratios and 95% confidence intervals for variables of interest, including *THEMIS2* expression. A *p*-value of less than 0.05 was considered statistically significant in all the analyses.

## 3. Results

### 3.1. THEMIS2 Expression in GBM Clinical Subgroups and Its Prognostic Association with Survival

We first analyzed the *THEMIS2* mRNA expression in GBM and LGG using the TCGA and GTEx databases via the GEPIA platform. *THEMIS2* was significantly upregulated in tumor tissues compared to normal brain tissues (Figure 1A). Next, we analyzed *THEMIS2* expression across different GBM molecular subtypes using the TCGA data. *THEMIS2* expression was significantly elevated in the mesenchymal subtype compared to the Classical and Proneural subtypes, with a moderate increase in the Neural subtype (Figure 1B). *THEMIS2* expression did not show significant differences based on the MGMT promoter methylation status (Figure 1C) and age groups (Figure 1D). Regarding the relationship between *THEMIS2* expression and MGMT promoter methylation status, we have performed survival analyses to address this aspect. These results have been added to the Appendix A. Specifically, in the CGGA dataset, we observed that a high *THEMIS2* expression was associated with poorer survival in patients with MGMT-methylated glioblastoma (Appendix A). However, no statistically significant survival differences were observed in the other subgroups (Appendix A). To assess the prognostic value of *THEMIS2*, a Kaplan–Meier survival analysis was performed. The results showed that patients with a high *THEMIS2* expression exhibited significantly shorter overall survival in both the TCGA (Figure 1E, *p* = 0.036) and CGGA cohorts (Figure 1F, *p* = 0.014).

### 3.2. Functional Enrichment of THEMIS2-Associated Genes

We next investigated the biological functions and pathways associated with *THEMIS2* in GBM using enrichment analyses from the TCGA and CGGA datasets. The GO and KEGG analyses were performed to explore the functional roles of *THEMIS2*-correlated genes. In the TCGA dataset, the GO enrichment analysis revealed that *THEMIS2*-correlated genes were significantly enriched in immune-related processes, such as “activation of immune response”, “leukocyte mediated immunity”, and “positive regulation of cytokine production” (Figure 2A). The KEGG pathway analysis supported these findings, revealing that *THEMIS2*-correlated genes were enriched in immune and inflammatory pathways, including “cytokine-cytokine receptor interaction”, “Th17 cell differentiation”, and “B cell receptor signaling pathway” (Figure 2B). GSEA further identified immune-related pathways, such as “Interferon Gamma Response”, “IL-6/JAK/STAT3 Signaling”, and “TNF-α Signaling via NF-κB” (Figure 2C). We observed similar results in the CGGA dataset. The GO enrichment analysis indicated that *THEMIS2*-correlated genes were significantly involved in immune processes, including the “positive regulation of cytokine production”, “leukocyte mediated immunity”, and “activation of immune response” (Figure 2D). The KEGG pathway analysis showed enrichment in pathways such as “cytokine-cytokine receptor interaction”, “lysosome”, and “B cell receptor signaling pathway” (Figure 2E). GSEA also identified immune-related pathways, including “inflammatory response”, “ Interferon Gamma Response”, and “TNF-α Signaling via NF-κB” (Figure 2F). These results suggest that *THEMIS2* is consistently involved in key immune-related processes and pathways, potentially contributing to the immune environment in GBM and affecting tumor progression and patient outcomes.

### 3.3. Immune Infiltration and THEMIS2 Correlation

We analyzed the correlation between *THEMIS2* expression and immune cell infiltration in GBM using the TIMER 2.0 database [24]. *THEMIS2* expression was significantly correlated with the infiltration levels of several immune cells, including neutrophils, B cells, macrophages, monocytes, and dendritic cells (Figure 3A). *THEMIS2* expression also showed a negative correlation with tumor purity (Rho = −0.584, *p* = 5.23 × 10^−14^), suggesting that higher *THEMIS2* expression may be associated with a more immune-enriched tumor microenvironment. We further examined immune cell infiltration using the MCP-counter and xCell algorithms. In the TCGA cohort, a high *THEMIS2* expression was associated with significantly increased infiltration of immune cells, such as neutrophils, NK cells, and macrophages (Figure 3B,C). Similarly, in the CGGA cohort, a high *THEMIS2* expression was linked to the elevated infiltration levels of monocytes, dendritic cells, and B cells (Appendix A). To explore the broader relationship between *THEMIS2* and immune cell infiltration, we applied the ssGSEA algorithm to assess the infiltration of 24 immune cell types. *THEMIS2* expression was positively correlated with the infiltration of most immune cell types in both the TCGA (Figure 3D) and CGGA datasets (Appendix A). These findings consistently indicate that *THEMIS2* is associated with increased immune cell infiltration, suggesting its significant role in shaping the immune landscape of the tumor microenvironment in glioblastoma.

### 3.4. Correlation Between THEMIS2 Expression, Immune Microenvironment, and Immune Checkpoint Molecules in GBM

We conducted the calculation of ESTIMATEScore, ImmuneScore, and StromalScore using the ESTIMATE algorithm to evaluate the distribution of immune and stromal components in the tumor microenvironment. The results showed a significant positive correlation between *THEMIS2* expression and ESTIMATEScore, ImmuneScore, and StromalScore in both the TCGA dataset (Figure 4A–C) and the CGGA dataset (Appendix A). A high *THEMIS2* expression was associated with elevated scores, suggesting that higher *THEMIS2* expression may reflect a more immune-enriched and stroma-enriched tumor microenvironment. Furthermore, we examined the correlation between *THEMIS2* expression and key immune checkpoint molecules. In both the TCGA and CGGA datasets, *THEMIS2* expression was positively correlated with multiple immune checkpoint molecules, including *HAVCR2*, *PDCD1LG2*, *VSIR*, *CSF1R*, *TREM2*, *IL10*, *IL10RA*, and *CD163*. (Figure 4D for TCGA and Appendix A for CGGA). This may contribute to the formation of an immunosuppressive microenvironment in glioblastoma, providing crucial insights into the potential role of *THEMIS2* in immune regulation.

### 3.5. Single-Cell Analysis of THEMIS2 Expression in GBM Microenvironment

We analyzed the single-cell RNA sequencing data to investigate the expression pattern of *THEMIS2* within the tumor microenvironment. Using UMAP dimensionality reduction, we visualized the distribution of various cell populations within the GBM samples, highlighting the cellular heterogeneity of the tumor (Figure 5A). *THEMIS2* was found to be primarily enriched in the macrophages, as indicated by the specific regions of the UMAP plot (Figure 5B). To further explore the expression of *THEMIS2* in the macrophages, we examined its expression within the macrophage cluster. The results demonstrated that *THEMIS2* was highly expressed in this immune cell population (Figure 5C). These findings suggest that *THEMIS2* plays a key role in macrophage activity and may influence the immune landscape of glioblastoma. The macrophage markers used for cell annotation are provided in the Appendix A.

### 3.6. Macrophage Subtype Analysis Based on THEMIS2 Expression

We divided the macrophages into two groups based on the median expression of *THEMIS2*, categorizing them as high- and low-expression groups (Figure 6A). Subsequently, we performed differential expression analyses (DEGs) between the high and low *THEMIS2*-expressing macrophages. The volcano plot in Figure 6B highlights the significantly differentially expressed genes between these two groups. Genes such as *CTSB*, *MSR1*, *GPR183*, and *IL10RA*, which are known to be associated with macrophage function and immune response, were significantly upregulated in the high-expression group. These DEGs were subsequently subjected to the GO enrichment analysis, revealing that the top enriched biological processes included phagocytosis, positive regulation of cytokine production, immune response-activating signaling pathways, and myeloid leukocyte activation (Figure 6C). Furthermore, gene network analysis highlighted the key pathways and genes associated with macrophage activation and immune response regulation, showing extensive interactions between the genes involved in phagocytosis, regulation of immune effector processes, and leukocyte activation (Figure 6D).

### 3.7. Intercellular Communication

To investigate the intercellular communication patterns associated with *THEMIS2* expression in the macrophages, we employed the “CellChat” package. The RNA-Seq data from the Seurat object was used to identify the overexpressed genes and predict ligand–receptor interactions across different cell types. Communication probabilities were calculated and aggregated at the signaling pathway level. The number of intercellular interactions between these cell populations is visualized in Figure 7A. The strength (or weight) of these interactions was also analyzed, with Figure 7B showcasing the intensity of communication between these cell types. Both figures show that the macrophages with high *THEMIS2* expression exhibited stronger interactions between cells. Lastly, the ligand–receptor pairs driving these intercellular communications are identified and visualized in Figure 7C. The identified pairs were linked to critical biological processes, including tumor migration and invasion promotion, immune suppression, immune cell regulation, immune escape, and enhancing tumor growth.

## 4. Discussion

Glioblastoma is one of the most aggressive primary brain tumors, characterized by significant intratumoral heterogeneity and an immunosuppressive tumor microenvironment, both of which contribute to therapy resistance and rapid disease progression. Our study identifies *THEMIS2* as a novel factor significantly overexpressed in GBM, particularly in the mesenchymal subtype, and predominantly expressed in macrophages. Its high expression is associated with poor overall survival. These findings are consistent with prior studies indicating that the mesenchymal subtype is linked to worse prognosis and increased immune activity within the TME [5,25,26].

Our findings suggest that a high THEMIS2 expression correlates with poorer survival in MGMT-methylated glioblastoma patients, potentially offsetting the survival advantage typically associated with MGMT methylation. This raises the possibility that suppressing *THEMIS2* expression could improve therapeutic outcomes, particularly in chemotherapy-sensitive patients. By mitigating the negative impact of *THEMIS2* on survival, the targeted inhibition of this gene might enhance the efficacy of alkylating agents such as temozolomide, thereby prolonging survival in this subgroup. These insights provide a foundation for future research aimed at refining treatment strategies for MGMT-methylated glioblastoma patients, offering a more individualized approach to optimize clinical outcomes.

Our results also demonstrate that *THEMIS2* expression is positively correlated with the infiltration of multiple immune cell types, including neutrophils, macrophages, dendritic cells, and T cells, suggesting a role in modulating the immune landscape within the TME. Moreover, a high *THEMIS2* expression was associated with increased levels of immune checkpoint molecules, such as *PD-1*, *PD-L1*, *CTLA-4*, and *HAVCR2*, indicating that *THEMIS2* might contribute to establishing an immunosuppressive TME that facilitates tumor immune escape. These findings align with the current understanding of the GBM immune landscape, characterized by immune evasion and suppression, which limits the efficacy of immunotherapies [27,28]. Single-cell RNA sequencing pinpointed macrophages as the primary cell type expressing *THEMIS2* within the TME, suggesting that macrophages may be key players in *THEMIS2*-mediated immune regulation in GBM. High *THEMIS2*-expressing macrophages exhibited the upregulation of the genes related to cytokine production, tumor promotion, immune suppression, and immune evasion. Given the crucial role of tumor-associated macrophages (TAMs) in promoting GBM progression and immune suppression, our findings suggest that *THEMIS2* may drive TAMs towards a pro-tumorigenic phenotype [29], supporting tumor growth and immune evasion.

The identified ligand–receptor pairs were linked to key biological processes, including tumor migration and invasion, immune suppression, and immune escape. These observations suggest that *THEMIS2* enhances pro-tumorigenic macrophage behavior through increased intercellular communication, promoting tumor progression and immune suppression. The positive correlation between *THEMIS2* expression and immune checkpoint molecules further suggests a role in regulating immune checkpoints, contributing to immune tolerance in the TME. This may explain the poorer prognosis observed in patients with high *THEMIS2* expression.

Therapeutically, targeting *THEMIS2* could modulate the immune response by altering TAM function, reducing immune suppression, and enhancing T-cell-mediated anti-tumor responses. Inhibiting *THEMIS2* may represent a novel strategy to reprogram TAMs from a pro-tumorigenic to an anti-tumorigenic phenotype, thereby modulating the TME to support effective anti-tumor immunity. Combining *THEMIS2*-targeted therapies with immune checkpoint inhibitors could potentially transform the immunologically “cold” TME of GBM into a “hot” one, increasing tumor immunogenicity and improving patient outcomes [2,30,31,32]. Such a combinatorial approach could address the limitations of immune checkpoint inhibitors in GBM, where the immunosuppressive TME poses significant challenges for effective immune activation.

Despite the promise of targeting *THEMIS2*, several challenges must be addressed. The heterogeneity of TAMs and their overlapping phenotypes with other myeloid cells necessitates precise targeting strategies to minimize off-target effects. Advanced single-cell technologies and high-throughput proteomics can facilitate the identification of specific markers and functional states of TAMs, enabling the development of more selective therapies. Further functional studies are required to elucidate the exact mechanisms by which *THEMIS2* influences macrophage behavior and immune cell infiltration. Additionally, in vivo models are needed to validate the therapeutic potential of *THEMIS2* inhibition and assess its impact on tumor progression and TME modulation.

## 5. Conclusions

Our study provides novel insights into the role of *THEMIS2* in GBM, highlighting its association with immune cell infiltration, macrophage function, and immune checkpoint regulation. *THEMIS2* emerges as a potential biomarker for prognosis and a candidate target for therapeutic intervention aimed at modulating the tumor immune microenvironment. Targeting *THEMIS2*, particularly in combination with ICIs, offers a promising strategy to enhance the efficacy of immunotherapies and improve outcomes for patients with GBM.

## Figures and Tables

**Figure 1 cells-14-00066-f001:**
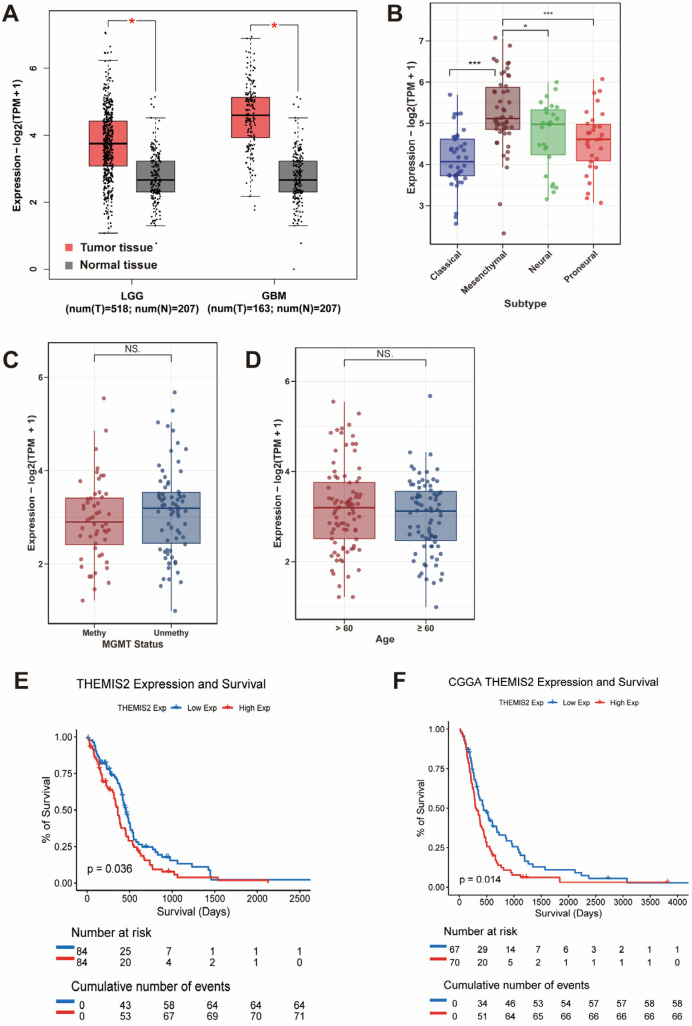
*THEMIS2* expression in GBM subgroups and survival. (**A**) *THEMIS2* mRNA expression in GBM and LGG compared to normal brain tissues (TCGA and GTEx via GEPIA); red bars represent tumor tissues (GBM and LGG), while gray bars represent normal brain tissues. (**B**) *THEMIS2* expression in GBM subtypes is higher in mesenchymal compared to Classical and Proneural, with moderate expression in Neural. (**C**,**D**) *THEMIS2* expression by MGMT promoter methylation and age. (**E**,**F**) Kaplan–Meier survival analysis showing shorter survival for high *THEMIS2* expression in TCGA and CGGA cohorts. * indicates *p* < 0.05, *** indicates *p* < 0.001, NS indicates not significant (*p* ≥ 0.05).

**Figure 2 cells-14-00066-f002:**
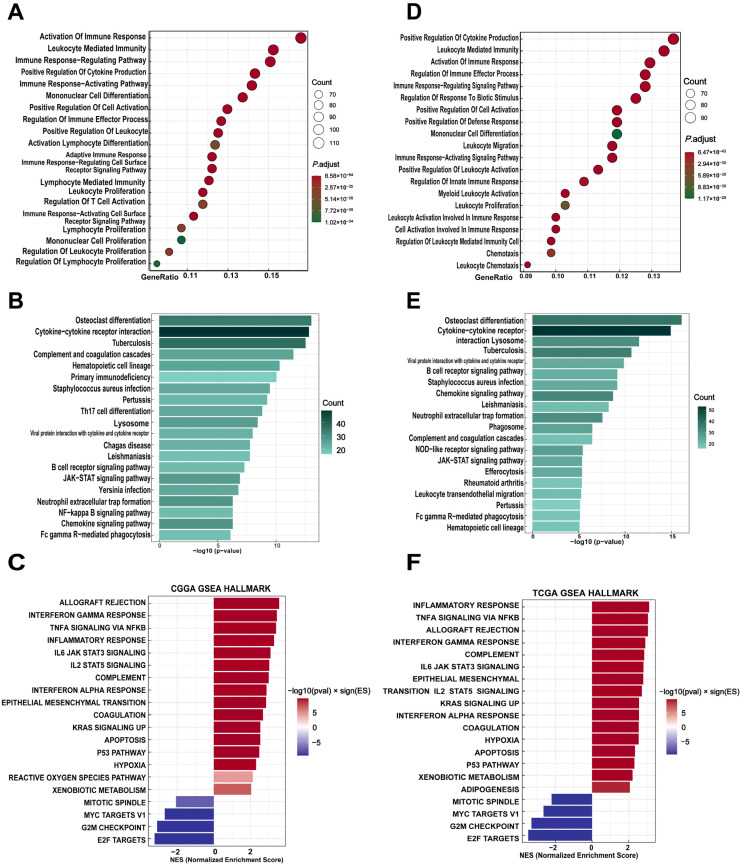
Functional enrichment of *THEMIS2*-associated genes. (**A**,**D**) GO enrichment analysis of *THEMIS2*-correlated genes in TCGA and CGGA, highlighting immune-related processes. (**B**,**E**) KEGG pathway analysis showing enrichment in immune and inflammatory pathways. (**C**,**F**) GSEA showing enrichment in immune-related pathways.

**Figure 3 cells-14-00066-f003:**
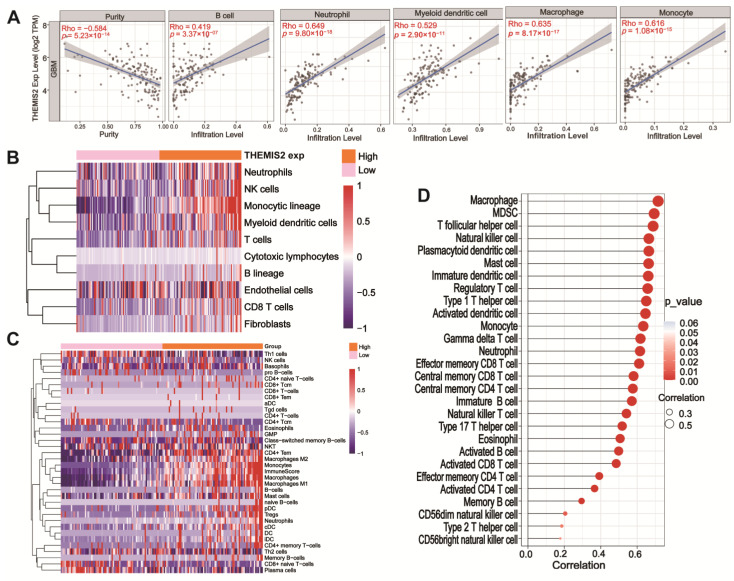
Immune infiltration and *THEMIS2* correlation. (**A**) Correlation between *THEMIS2* expression and immune cell infiltration using TIMER 2.0, showing associations with tumor purity, B cells, neutrophils, myeloid dendritic cells, macrophages, and monocytes. (**B**,**C**) MCP-counter and xCell analysis showing increased immune cell infiltration with high *THEMIS2* in TCGA dataset. (**D**) ssGSEA analysis showing positive correlation with immune cell infiltration in TCGA dataset.

**Figure 4 cells-14-00066-f004:**
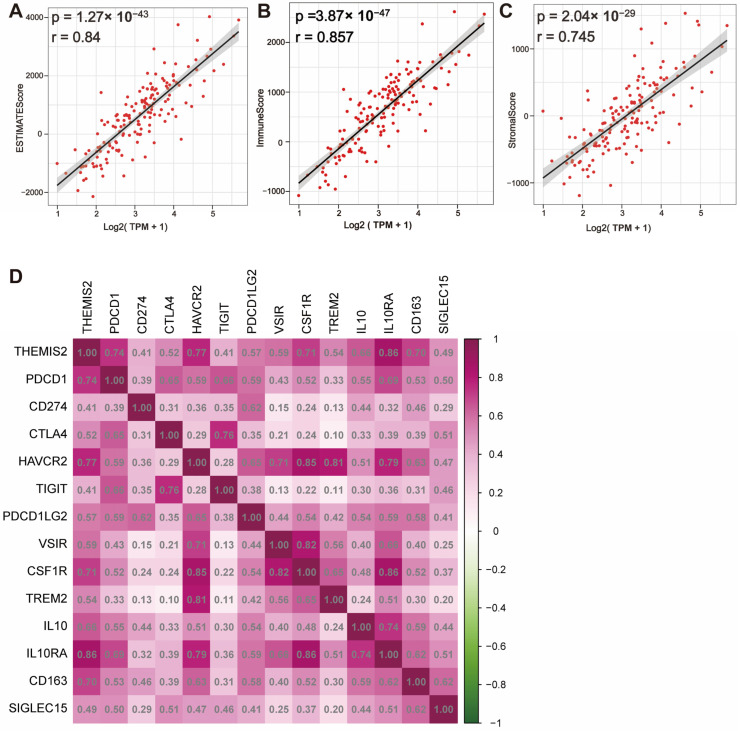
*THEMIS2* expression, immune microenvironment, and immune checkpoints. (**A**–**C**) Correlation of *THEMIS2* with ESTIMATEScore, ImmuneScore, and StromalScore in TCGA, indicating positive correlations. (**D**) Correlation of *THEMIS2* with common immune checkpoint molecules in TCGA, suggesting a role in an immunosuppressive microenvironment.

**Figure 5 cells-14-00066-f005:**
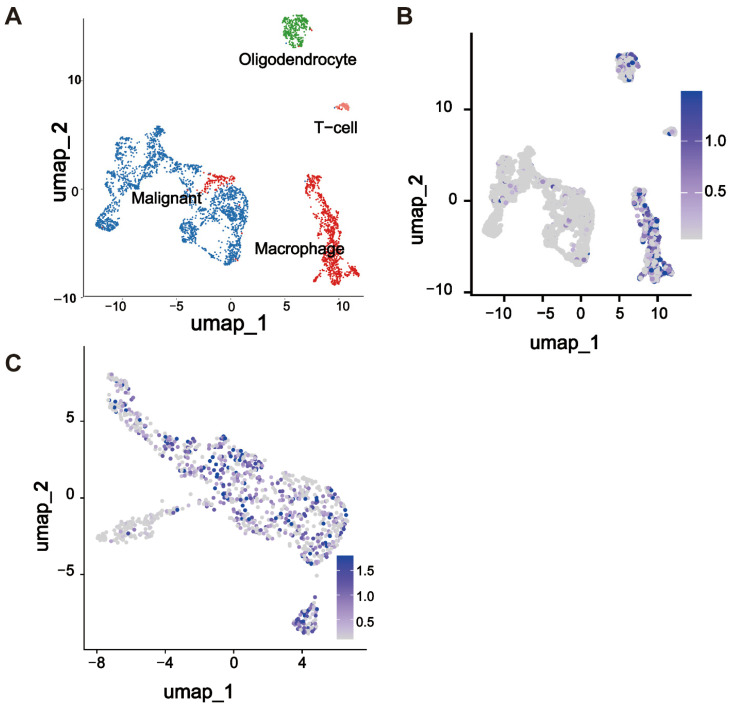
Single-cell analysis of *THEMIS2* in GBM. (**A**) UMAP plot of GBM single-cell RNA sequencing showing the cell population distribution. (**B**) *THEMIS2* expression enriched in macrophages. (**C**) High *THEMIS2* expression within the macrophage cluster.

**Figure 6 cells-14-00066-f006:**
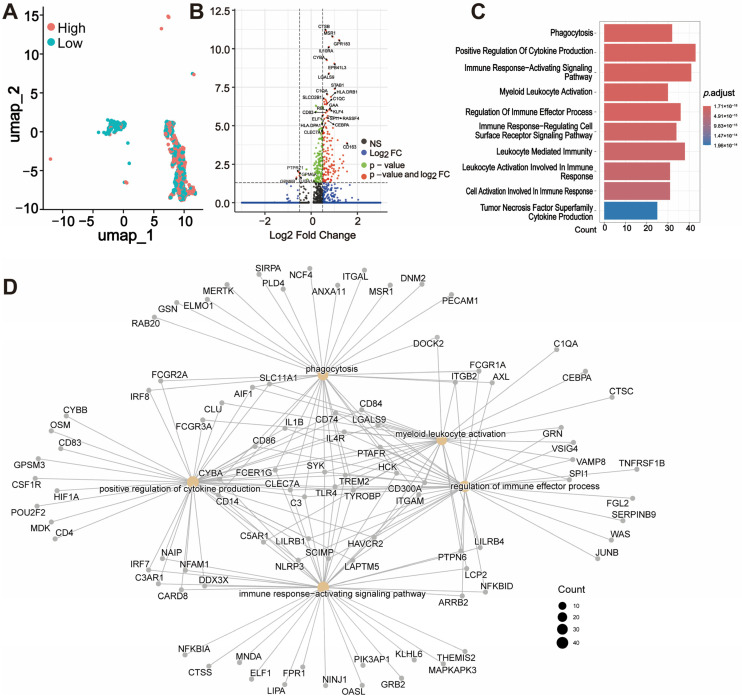
Macrophage subtype analysis based on *THEMIS2*. (**A**) Division of macrophages by high and low *THEMIS2* expression. (**B**) Volcano plot of DEGs between high and low *THEMIS2* groups. (**C**) GO enrichment of DEGs showing phagocytosis and immune activation. (**D**) Gene network analysis of macrophage activation and immune response.

**Figure 7 cells-14-00066-f007:**
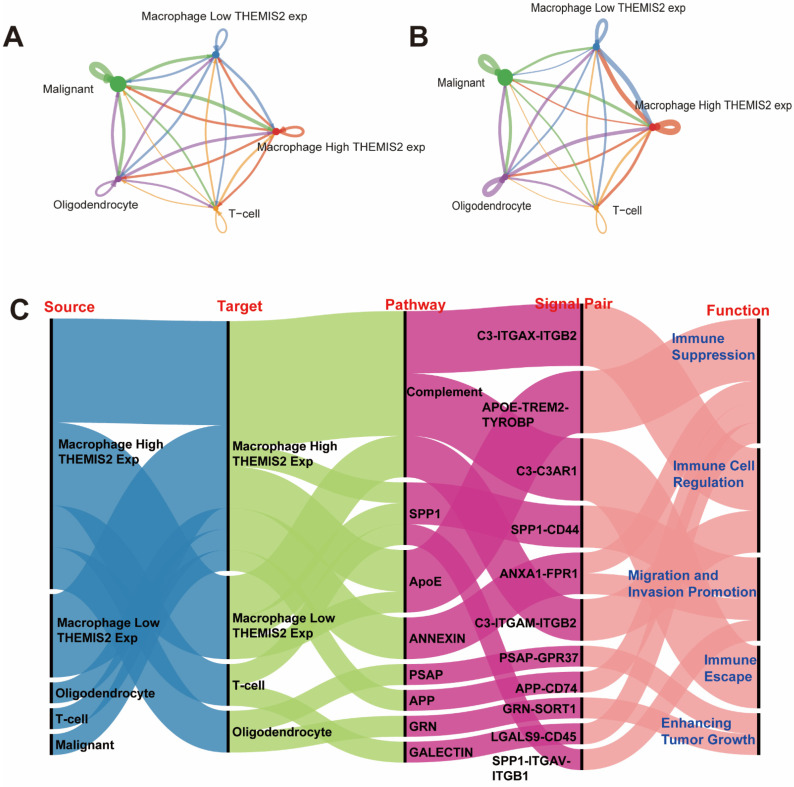
Intercellular communication analysis. (**A**) Number of intercellular interactions using CellChat showing stronger communication in high *THEMIS2* macrophages. (**B**) Interaction strength analysis. (**C**) Ligand–receptor pairs linked to tumor progression and immune modulation. (**A**,**B**) The colored lines in the network represent communication probabilities between cell types, with warmer colors (e.g., red) indicating stronger interactions and cooler colors (e.g., blue) indicating weaker interactions. Line thickness corresponds to the strength of the communication.

## Data Availability

The data supporting the findings of this study are available upon reasonable request from the corresponding author. The R code used in this study will be made available upon reasonable request after publication. The authors acknowledge the use of ChatGPT for refining grammar and improving the readability of the manuscript.

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
