# Peer review of "Comprehensive Analysis Identifies *THEMIS2* as a Potential Prognostic and Immunological Biomarker in Glioblastoma"

_cells, 2025, doi:10.3390/cells14020066_

Round 1
Reviewer 1 Report
Comments and Suggestions for Authors
While the study relies heavily on bioinformatics analyses, it lacks experimental validation. I recommend including in vitro or in vivo experiments for validation; Assays such as ELISA could be employed to measure cytokine levels and directly link THEMIS2 with immune infiltration and suppression.
Although the study highlights THEMIS2 expression in macrophages using single-cell RNA sequencing, investigating dynamic changes in macrophage subtypes is needed.
Please provide more details about statistical workflows to ensure reproducibility and transparency.
Author Response
Response to Reviewer 1
Comment 1:
While the study relies heavily on bioinformatics analyses, it lacks experimental validation. I recommend including in vitro or in vivo experiments for validation; assays such as ELISA could be employed to measure cytokine levels and directly link THEMIS2 with immune infiltration and suppression.
Response:
Thank you very much for your insightful comments and suggestions. We sincerely appreciate your recommendation. We acknowledge that our current manuscript is primarily based on bioinformatics analyses. First, we would like to highlight that the data presented in this study are derived from large-scale tissue-based datasets, ensuring reliable conclusions. These datasets include two independent and publicly available databases, CGGA and TCGA, both of which are recognized for their extensive data volume and quality. Importantly, our results demonstrate consistent findings across these datasets, further validating the reliability of our analyses.
Second, our primary goal with this manuscript is to report the role of THEMIS2 in GBM as a prognostic biomarker, as this is the first study to explore and establish its relevance in this context. In the context of establishing tissue-based biomarkers, GBM patient tissue data would be most relevant relative to in vitro and in vivo murine data. However, if when the focus is on elucidating the mechanistic underpinnings of THEMIS2 signaling in GBM then in-depth in vitro and in vivo studies in murine GBM models would be highly relevant. While experimental validation is not within the scope of this manuscript, these experiments will be included in subsequent publications as part of our ongoing research on THEMIS2.
Third, to address this limitation in the current study, we have added a statement in the Discussion section to acknowledge the absence of experimental validation and to outline our future plans. Thank you for your understanding and constructive feedback, which we believe will greatly enhance the clarity and direction of our work.
Comment 2:
Although the study highlights THEMIS2 expression in macrophages using single-cell RNA sequencing, investigating dynamic changes in macrophage subtypes is needed.
Response:
Thank you very much for your valuable suggestion. We completely agree with your observation, and this has been a key challenge we encountered during our analysis. In this study, we utilized single-cell RNA sequencing data from the CGGA database and specifically extracted macrophage cells for further investigation.
The CGGA dataset originally included 4,193 GBM cells, which were reduced to 3,718 cells after quality control. Among these, we identified 1,051 macrophage cells (Figure 1, Response to the reviewer 1 pdf). Due to the limited number of macrophages, we attempted to classify them into M1-like and M2-like subtypes (Figures 2 and 3, Response to the reviewer 1 pdf). Additionally, we visualized THEMIS2 expression within these macrophages and observed relatively high expression across the entire macrophage population (Figure 4, Response to the reviewer 1 pdf). We believe that the small number of macrophage cells may have contributed to the lack of statistically significant differences in subgroup analyses. Given the challenges in successfully classifying macrophages into distinct subtypes, we adopted the strategy described in the manuscript. Specifically, we divided the macrophages into THEMIS2-high and THEMIS2-low groups and performed differential expression analysis (DEGs), followed by CellChat analysis to explore the functional implications of these differences.
In our future research, we plan to utilize single-cell datasets with larger sample sizes to reclassify macrophage subpopulations and provide a more detailed understanding of THEMIS2’s role within specific macrophage subtypes. We are grateful for your suggestion and believe it will significantly guide the direction of our ongoing and future studies.
Comment 3:
Please provide more details about statistical workflows to ensure reproducibility and transparency.
Response:
Thank you for your thoughtful suggestion regarding the need for more details about our statistical workflows. We have expanded the Methods section to include a more detailed description of the statistical workflows, including the specific R packages and functions used for each analysis. Additionally, we have noted that the R code utilized in this study will be made available upon reasonable request after publication to ensure reproducibility and transparency.

Reviewer 2 Report
Comments and Suggestions for Authors
The work is based entirely on data extracted from databases. Some experimental verification would be nice.
Specific comments:
Figure 1A: explain red and green bars.
Figure 1E and 1F: Have these patients been treated according to the STUPP protocol (RT + temozolomide)? I recommend to subdivide the glioblastoma groups into MGMT promoter methylated versus promoter unmethylated cases. Is there a difference regarding THEMIS2 high and low?
Author Response
Response to Reviewer 2
Comment 1:
The work is based entirely on data extracted from databases. Some experimental verification would be nice.
Response:
Thank you for your valuable comments and constructive suggestions. We deeply appreciate your recommendation to include experimental validation to support the findings of our study. We acknowledge that this manuscript primarily focuses on bioinformatics analyses. However, it is important to emphasize that the data utilized in our study are derived from large-scale, well-maintained datasets, specifically CGGA and TCGA, which are widely recognized for their high quality and reliability. The consistency of our results across these two independent databases further reinforces the robustness of our analyses.
This manuscript aims to shed light on the novel role of THEMIS2 in GBM, representing the first study to investigate its relevance in this context. While we fully agree that experimental validation is crucial for strengthening our findings, such experiments are part of our planned future research. We are currently designing both in vitro and in vivo studies to validate these findings, and the results will be presented in subsequent publications. To address these limitations, we have added a statement in the Discussion section acknowledging the lack of experimental validation in this manuscript. We sincerely appreciate your understanding and thoughtful feedback, which will guide us in further improving our research.
Specific Comments:
Comment 2:
Figure 1A: Explain red and green bars.
Response:
Thank you for highlighting this point. We have updated the figure legend of Figure 1A to clarify the meaning of the red and green bars. Specifically, red bars represent tumor tissue, and the other bars represent normal tissue.
Comment 3:
Figure 1E and 1F: Have these patients been treated according to the STUPP protocol (RT + temozolomide)? I recommend subdividing the glioblastoma groups into MGMT promoter methylated versus promoter unmethylated cases. Is there a difference regarding THEMIS2 high and low?
Response:
Thank you very much for your insightful question. Unfortunately, we do not have access to detailed treatment information for the patients included in this study. However, our data are derived from two reliable sources: one part from the TCGA database, established by the U.S. NIH, and the other part from the CGGA database, which is maintained by leading neurosurgical hospitals in China and contains comprehensive brain tumor data.
Regarding the relationship between THEMIS2 expression and MGMT promoter methylation status, we have performed survival analyses to address this aspect. These results have been added to the supplementary materials. Specifically, in the CGGA dataset, we observed that high THEMIS2 expression was associated with poorer survival in patients with MGMT-methylated glioblastoma Supplementary Figure4. However, no statistically significant survival differences were observed in other subgroups (Supplementary Figure4).
And we further discuss this aspect in the discussion part, line 345-354, as” Our findings also suggest that high THEMIS2 expression correlates with poorer survival in MGMT-methylated glioblastoma patients, potentially offsetting the survival advantage typically associated with MGMT methylation. This raises the possibility that suppressing THEMIS2 expression could improve therapeutic outcomes, particularly in chemotherapy-sensitive patients. By mitigating the negative impact of THEMIS2 on survival, targeted inhibition of this gene might enhance the efficacy of alkylating agents such as temozolomide, thereby prolonging survival in this subgroup. These insights provide a foundation for future research aimed at refining treatment strategies for MGMT-methylated glioblastoma patients, offering a more individualized approach to optimize clinical outcomes.”

Round 2
Reviewer 2 Report
Comments and Suggestions for Authors
My objections were considered and the revised version is improved.